# The Impact of the COVID-19 Pandemic on the Global Web and Video Conferencing SaaS Market

Cristiana Tudor

International Business and Economics Department, Bucharest University of Economic Studies, 010374 Bucharest, Romania; cristiana.tudor@net.ase.ro

**Abstract:** The COVID-19 pandemic related government interventions produced rapid decreases in worldwide economic and social activity, with multifaceted economic and social consequences. In particular, the disruption of key industries and significant lifestyle changes in the aftermath of the pandemic outbreak led to the exponential adoption of web and video conferencing Software as a Service (SaaS) programs and to the solutions-led video conferencing market growth. However, the magnitude and persistence of the COVID-19 pandemic impact on the video conferencing solutions segment remain uninvestigated. Building on previous evidence linking population web-search behavior, private consumption, and retail sales, this study sources and employs Google Trends data as an analytical and forecasting tool for the solutions segment of the videoconferencing market. It implements a univariate forecast evaluation approach that assesses the predictive performance of several statistical and machine-learning models for the relative search volume (RSV) in the two SaaS program leaders, Zoom and Teams. ETS is found to provide the best forecast of consumer GT search interest for both RSV series. A baseline level for the consumer interest over the first pandemic wave is subsequently produced with ETS and further serves to estimate the excess search interest over the February 2020–August 2020 period. Results indicate that the pandemic has created an excess or abnormal consumer interest in the global web and videoconferencing SaaS market that would not have occurred in the absence of the pandemic. Other findings indicate that the impact is persistent as the excess interest stabilized at higher levels than in the pre-pandemic period for both SaaS market leaders, although a higher saturation of the Zoom market is detected.

**Keywords:** ARIMA; big data analytics; COVID-19 impact; Google Trends; ETS; forecast; neural networks; software as a service (SaaS)

## 1. Introduction

The global spread of the Severe Acute Respiratory Syndrome Coronavirus (SARS-CoV-2), ordinarily referred to as Coronavirus Disease 2019 (COVID-19) [1] in early 2020 was an unprecedented and highly disruptive event [2]. The World Health Organization (WHO) defined COVID-19 as a pandemic on 11 March 2020 [3], which in turn led to a broad spectrum of government interventions to stop the disease's spread that were unparalleled in recent history, including, inter alia, social distancing measures, lockdowns, travel restrictions, and the closure of schools and nonessential businesses [4–8]. These measures led to rapid decreases in economic and social activity, with far-ranging economic consequences [9], making the coronavirus pandemic considerably more than a medical emergency ([1,10]). The COVID-19 pandemic affected the USD 90 trillion global economy [11] and induced the sharpest economic contraction since the Great Depression in the 1930s, with estimations indicating losses of USD 8.5 trillion of the global economic output [12]. Consequently, the COVID-19-induced crisis can be characterized as a black swan event [13] with multifaceted economic and social consequences [14].

The coronavirus pandemic has disrupted schooling in over 150 nations, impacting 1.6 billion children [15]. Consequently, the education response concentrated on remote

learning modalities as an emergency intervention [15] and the traditional educational methods were replaced by e-learning, i.e., a formal learning system that uses electronic resources [16–18]. In this circumstance, the use of video conferencing platforms as e-learning tools has increased significantly after the COVID-19 outbreak in early 2020 [17]. It should be mentioned that even before the pandemic, educational technology was seeing rapid expansion and adoption, with worldwide edtech investments totaling USD 18.66 billion in 2019 and the whole online education market anticipated to reach USD 350 billion by 2025 [19]. Moreover, with lockdown measures imposed globally, video conferencing applications (apps) or Software as a Service (SaaS) programs such as Zoom, Microsoft Teams, WebEx, and Google Meet have emerged as the ultimate solution for businesses and government organizations to connect with remote workers, consumers, and employees [20]. Furthermore, videoconferencing has also facilitated remote access to healthcare providers and diagnostic services during the COVID-19 pandemic [21]. These are some of the key factors that have accelerated the use of videoconferencing apps, which in turn has produced significant and lasting effects on the video conferencing market that has witnessed remarkable growth from 2020 [22]. Of note, on March 23, Zoom was downloaded 2.13 million times at the global level, whereas the app had just under 56,000 global downloads in a day two months before [23]. Estimations further indicate that the video conferencing market will continue to grow from USD 6.87 billion in 2022 at a CAGR of 11.3% to USD 14.58 billion by 2029 [24]. The video conferencing market is divided into two components: solutions and services. The highest market share is attributed to the solutions segment, which includes video conferencing software as a service (Saas) programs or apps and is responsible for the overall video conferencing market growth in the aftermath of the pandemic outbreak due to the exponential adoption of video conferencing SaaS among various industries [22], such as IT & communications, education, and healthcare. Furthermore, new technologies such as IoT, cloud computing, VR, enhanced video compression, and AI are expected to propel the video conferencing market [25].

In light of the above considerations, a quantitative assessment of the magnitude of the COVID-19 pandemic impact on the video conferencing solutions segment and, in particular, the persistence of this impact are timely and informative research topics for policymakers and market players.

This study aims to assess this impact by using Google Trends data in the form of relative search volume (RSV) time series to extract relevant information on variation in consumer preferences that is hard to detect otherwise. Google holds an impressive 86.19 percent of the search engine market as of December 2021 [26] and can thus identify trends in consumer behavior based on data analytics on web search activity [27]. Google Trends thus offer relevant and timely information on the relative frequency of search queries over time [28]. In particular, Google Trends data has been used in a wide range of research areas, from IT, communications, medicine, and health to business and economics [29]. The current research relies specifically on prior studies that recognized the utility of online search information in predicting private consumption and retail sales ([27,30–35]; [36–39]). Ref. [40] further show that demand predictions based on GT search queries are, on average, highly accurate approximations of reality.

As a result, it is acceptable to assume that a rise in web searches for a videoconferencing app reflects an increase in consumer interest and subsequent sales for that particular videoconferencing solution. Figure 1 reflects the evolution of Google Trends web queries for the main videoconferencing solutions over 2017–2022, confirming that GT can accurately mirror the evolution of the videoconferencing market over the same period. The chart also illustrates that, except for Zoom, Teams, and Meet, the other videoconferencing apps have not seen a dramatic boost in online interest in the aftermath of the COVID-19 pandemic outbreak in early 2020.

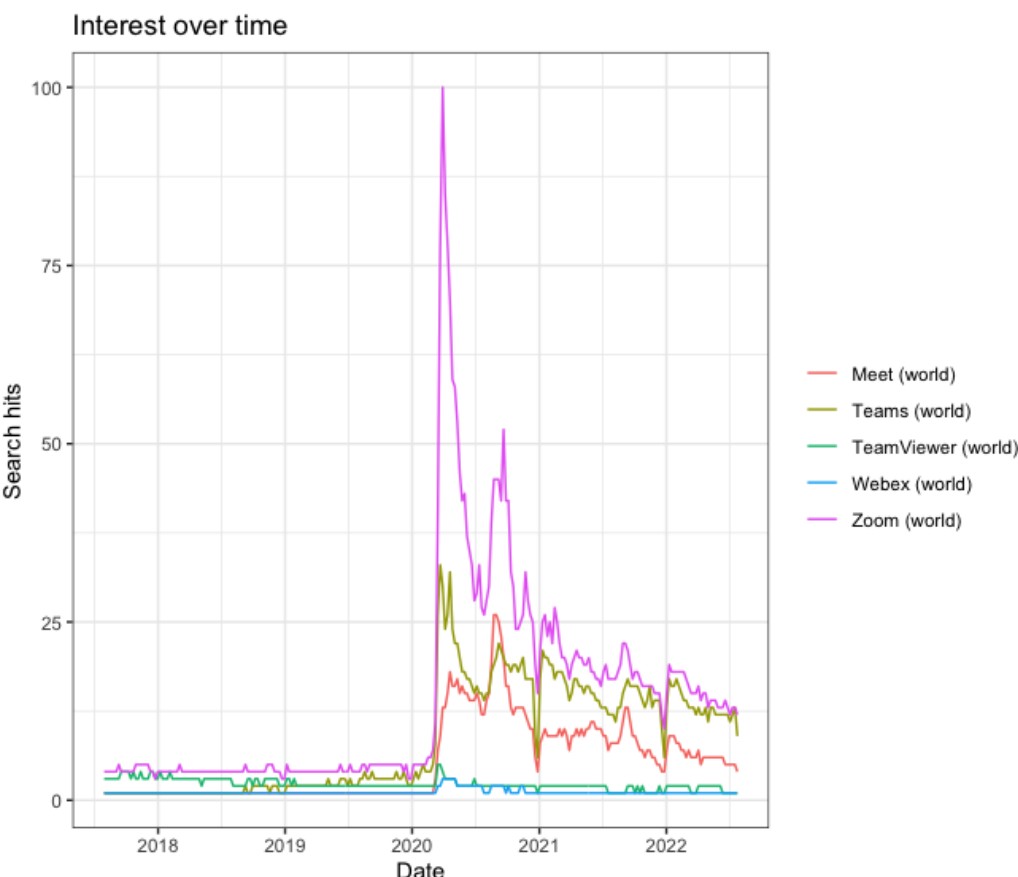

**Figure 1.** GT query for web and videoconferencing solutions (SaaS programs) (2017–2022); Created with the ggplot2 package in R environment with GT data sourced with the gtrendsR package.

In the context of the current research, "excess" search interest over the pandemic's early stages denotes an increase in online search activity for the particular solution over what would have occurred during "normal" conditions. Consequently, to detect the excess web search interest for videoconferencing apps at the global level, a baseline level that proxies for "normal" expected search activity and serves for comparative purposes is needed.

However, time series forecasting remains challenging, whereas the emergence of big data further complicates the issue [41]. Most previous studies that incorporate GT data in forecasting endeavors incorporate Google Trends search information into multivariate models and show that the inclusion of GT information offers significant benefits in the form of improved forecasting performance. Nonetheless, GT data has also been acknowledged in previous research as a leading indicator for key variables of interest [27]. In this context, univariate modeling and forecasting that "lets the data speak for itself" [42] emerges as the most suitable technique. Ref. [27] have previously used Burberry as an example to test whether there is a single univariate forecasting model that can correctly estimate fashion consumer Google Trends over various horizons. Consequently, both parametric and nonparametric forecasting models, including ARIMA, ETS, TBATS, and NNAR, were estimated, and results indicated that no single univariate model could outperform in forecasting the GT search index, whereas the NNAR model was identified as the worst performer. Ref. [43] also employed GT time series in a univariate setting to predict cancer incidence and cancer mortality. The study finds that the NNAR model can perform best in the out-of-sample context to predict web-query volume, surpassing ARIMA and TBATS in terms of forecasting ability. Ref. [26] explore whether there has been a rise in the number of "headache" searches on the Internet since the COVID-19 pandemic breakout, which could in turn indicate a higher prevalence of the health problem. To this end, several statistical and machine-learning methods (ARIMA, Holt-Winters, ETS, STS, TBATS, and NNAR) have

been estimated and their forecasting accuracy assessed on pre-pandemic testing sets, with results indicating that the Holt-Winters model is overperforming in predicting web query trends out-of-sample.

Thus, as the literature concerned with forecasting time series of the GT relative search volume index remains thin and no model has emerged as best-performing in a univariate framework, this study employs several statistical and machine-learning models (as per the terminology of [44]) including the innovations state-space models for exponential smoothing (ETS), the autoregressive integrated moving average (ARIMA) model, and the feed-forward neural network autoregression (NNAR) model, along with a naive model, to model and forecast the consumer interest for SaaS programs as it is reflected in web queries submitted to Google. The ultimate research goal is to assess the impact of the COVID-19 pandemic on the solutions segment of the videoconferencing market by making use of the produced forecast as a baseline level for the estimation of the excess consumer interest.

The study offers three main contributions to the extant literature in the form of (i) a robust approach for forecasting the relative search volume (RSV) that employs statistical and machine-learning prediction models; (ii) a quantitative assessment of excess search interest for videoconference solutions during the first pandemic wave; and (iii) new findings on the usefulness of big data in the form of Google Trends for predicting sales. Moreover, research findings can provide useful information to the videoconferencing solutions market players, such as identification of market saturation and the need for diversification, identifying key competitors, or revealing the success of implementing product enhancements. To the best of our knowledge, this is the first study to capitalize on the usefulness of Google Trends as an analytical and forecasting tool for the videoconferencing market.

Results indicate that the pandemic has resulted in excess consumer interest for video conferencing SaaS programs that wouldn't have emerged otherwise. Although the SaaS market began losing momentum in the second half of 2020, the excess consumer interest stabilized at higher levels relative to the pre-pandemic period, suggesting that the impact is persistent. Moreover, the decrease in consumer interest that is detected for both of the main SaaS programs, Zoom and Teams, indicates that the market is oversaturated and reveals that the main market players, especially Zoom, are under intense pressure to diversify their product portfolio. For Zoom, expansions into the contact center market and the hardware segment are possible avenues to counter the oversaturation of the core Meetings video-conferencing product.

The remainder of the study is organized as follows. Section 2 presents the data sample, the data splitting rules applied in the empirical investigation, and analyses trends in the consumers' interest in videoconferencing SaaS programs. It also presents the forecasting models, the accuracy measures, and the automatic forecasting technique. Section 3 contains the empirical results and tests for robustness. Section 4 discusses the research findings and includes the breakdown of the SaaS RSV by region for the US market. Section 5 concludes the study.

## 2. Materials and Methods

### 2.1. Data

2.1.1. Sample

Google Trends is a platform that reports the popularity of a keyword in a certain region over a specified time period. Specifically, it reports the index of the relative search volume (RSV) calculated as per Equation (1), which is then scaled from 0 to 100 [26]. Consequently, an RSV of 100 reflects the point of peak popularity for the particular term [27].

$$\text{RSV} = \frac{\text{Total query volume for search term in a given period and geography}}{\text{Total number of queries in that period and geography}} \tag{1}$$

For the quantitative investigation of the COVID-19 pandemic impact on the market for videoconferencing solutions, we sourced and analyzed GT data for the two main videoconferencing apps, Zoom (Zoom Communication) and Teams (Microsoft). Thus, we extracted the time series of RSV for the keywords "Zoom" and "Teams" for a five-year period spanning 30 July 2017–24 July 2022 at the world level, by selecting the category "Software" and the geography "World". This approach assures the consistency and relevancy of results, which is further reinforced by analyzing the related queries (available on request) for the two keywords. Each data series contains 261 weekly observations.

2.1.2. Data Splitting

However, to investigate the pandemic's impact on query trends, the following data splitting strategy is implemented. First, for each series, two sub-samples are subtracted, corresponding to the pre-pandemic period and the pandemic period, respectively. The starting point of the pandemic is set to be the week ending 26 January 2020, based on the fact that the World Health Organization declared the coronavirus outbreak a Public Health Emergency of International Concern on 31 January 2020. Second, the pre-pandemic window is further divided into a training set of length $l_T = 100$ weekly observations (on which various statistical and machine-learning models are fitted and optimized) and an (out-of-sample) testing set of length $l_t = 31$ weekly observations that serves to comparatively assess the forecasting accuracy of the optimized model specifications and identify the best performing forecasting method for each time series. Finally, a forecasting horizon *h* comprising 27 weeks spanning 27.01.2020 to 2.08.2020 is subset from the pandemic window. This is used to estimate the baseline level, which is produced by the over-performing predictive model identified in the testing set and corresponds to the "expected" web query activity under normal conditions or in the absence of the disruptive pandemic. The choice of the length of *h* is based on two important criteria: (i) it should be smaller than the testing window [45] and (ii) it should be short enough so as to minimize the risk of losing the forecasting accuracy that is bared by longer forecasts [46]. Of note, the best performing model specifications uncovered in the out-of-sample setting are first reestimated and their parameters optimized over the entire pre-pandemic window of length 131 before issuing point forecasts for the following 27 weeks.

Figure 2 offers a visual representation of the implemented data splitting strategy.

Then, as per Equation (2), the estimated baseline level is extracted from the actual web-search interest as reflected by RSV levels over the forecasting horizon for each time series to produce a quantitative measure of the impact that the COVID-19 pandemic has had on the videoconferencing solutions market, i.e., the excess RSV.

$$y_{i_h} = RSV_{i_h} - \hat{Y}_{i_h} \tag{2}$$

where $RSV_{i_h}$ is the actual RSV level for the keyword *i* at time *h* and $\hat{Y}_{i_h}$ is the forecasted baseline level for keyword *i* at time *h* under normal circumstances. An alternate method employed in previous studies uses averages of previous data over a recent period to represent the baseline level. However, as [26] argue, given data with trend or seasonality, as in this case, this strategy would underestimate expectations and consequently overstate excess web-search interest. As a result, using a credible forecasting model to predict the level of web-queries is a superior method.

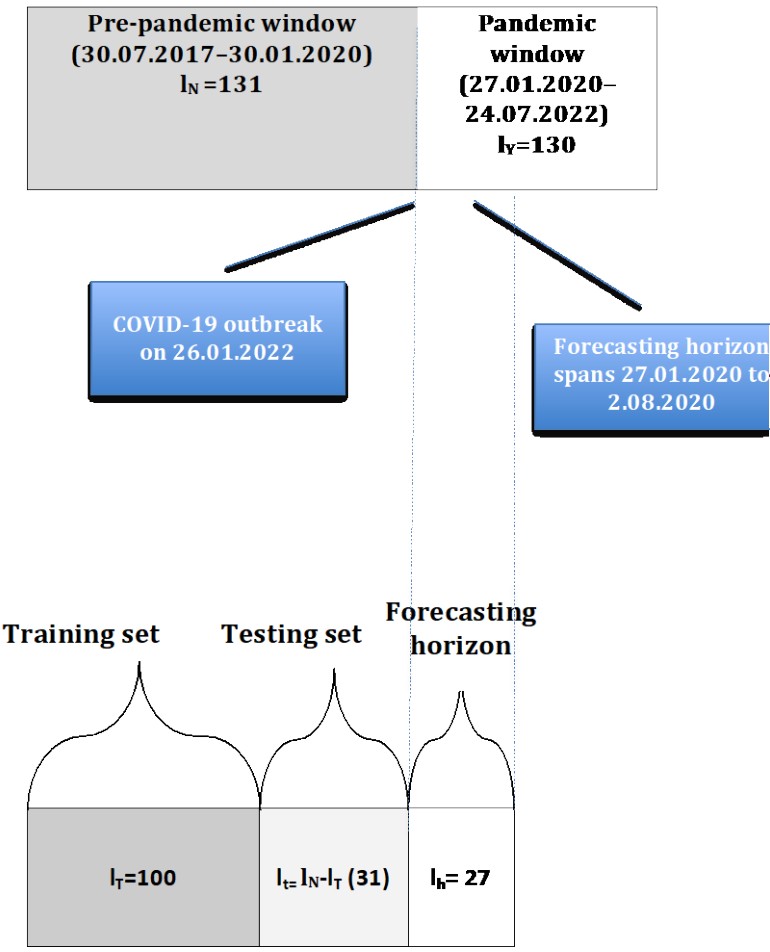

**Figure 2.** Data splitting strategy.

2.1.3. Trends in Search Interest for the Main Web and Videoconferencing SaaS Programs

Figure 3 plots the RSV time series for "Zoom" search interest as reflected by Google Trends data. For more robust and consistent findings, we delineate the "Software" category when sourcing GT data. Of note, the dedicated R package gtrendsR [47] is used to extract the data from Google Trends (www.google.com/trends (accessed on 27 July 2022)). Even if the language setting would not constitute a problem when extracting GT trends directly from the source given the current keywords, it is important to note that with the gtrendsR package, the language setting is only influencing the data returned by related topics, which is not a relevant factor in the current research. Another advantage of using the dedicated R package consists of the fact that it allows for a reproducible workflow.

The visual inspection reveals a low, constant interest in the pre-pandemic period and a significant surge by the end of March 2020, amid the first lockdowns imposed in the aftermath of WHO declaring the COVID-19 pandemic on 11 March 2020. The chart highlights that the peak popularity of the term "Zoom" was reached in the week ending 29 March 2020. The search interest declined thereafter until stabilizing by 2021 at a higher level relative to its pre-pandemic evolution. The lower part of the chart zooms over the first pandemic year while adding a smooth curve computed with the "loess" method within the "geom_smooth" function in the "ggplot2" package of the R environment. LOESS, also known as locally weighted polynomial regression, is a non-parametric approach that fits multiple regressions in a local neighborhood and has been proposed by [48]) and further developed by [49]. Loess smoothing facilitates the visualization of the trend of the data series and is a helpful tool in data analytics.

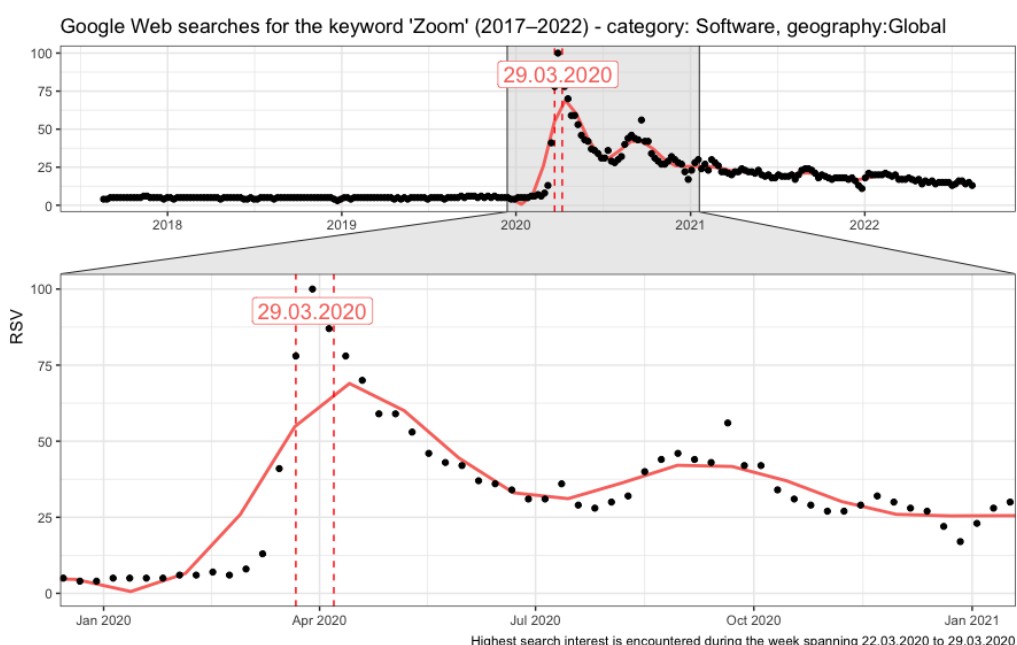

**Figure 3.** RSV for the keyword "Zoom", Category "Software", Geography "World" (July 2017–July 2022) (above); Zoom in over 2020; Loess smoothing is applied (below); Created with the ggplot2 package in R environment with GT data sourced with the gtrendsR package.

The visual inspection of the trend of search queries for the Microsoft Teams videoconferencing app, reflected in Figure 4, reveals a roughly similar evolution. However, some differences are also revealed, as for Teams, the peak popularity was attained one week prior, whereas the subsequent evolution of the RSV time series shows a smoother decreasing trend that stabilizes at significantly higher levels relative to its pre-pandemic levels and also at higher levels than in the case of the search interest for Zoom.

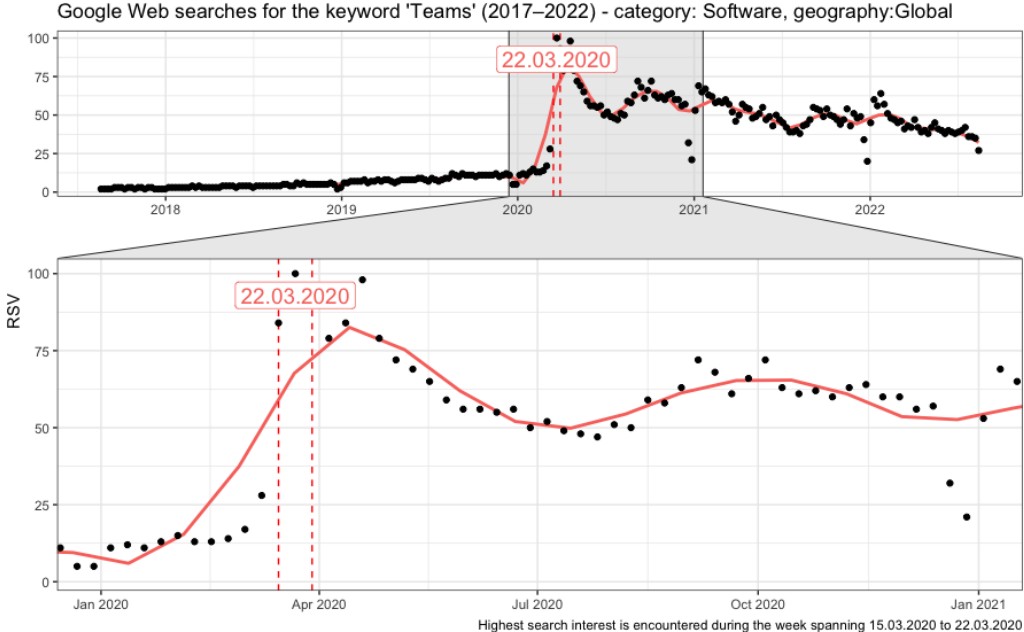

**Figure 4.** RSV for the keyword "Teams", Category "Software", Geography "World" (July 2017–July 2022) (above); Zoom in over 2020; Loess smoothing is applied (below); Created with the ggplot2 package in R environment with GT data sourced with the gtrendsR package.

*2.2. Method*

2.2.1. Forecasting Models

The exponential smoothing State Space Model (ETS) is built on the work of [50–52]. Forecasts issued by ETS are weighted averages of previous observations; there are higher weights applied to recent observations and overall exponentially declining weights ([45]; [27]). Three components are included in ETS estimations, namely the trend (T), seasonal (S), and error (E) components. Further, the trend combines a level term (l) and a growth term (b). As per [53], both the trend and seasonal components may be one of none (N), additive (A), additive damped (Ad), multiplicative (M), or multiplicative damped (Md). Thus, an ETS model is written as a three-character string (Z,Z,Z) [54], with the first Z denoting the error assumption of the state-space model, and the second and third Zs reflecting the type of the trend, and of the season. The implementation of the automated ETS algorithm via the ets function found within the forecast package in R [55,56] evaluates in the modeling process 30 ETS formulae (i.e., all 30 formulae are provided by [55]). The algorithm optimizes the smoothing parameters and the initial state variable and uses a penalized likelihood, i.e., the corrected Akaike's Information Criterion (AICc), to identify the best of the models on the training set, which is then employed to produce point forecasts.

Autoregressive integrated moving average (ARIMA) models have been developed by [57] and are one of the most popular parametric time series analysis and forecasting models [27]. In equation form, as per [45], a seasonal model ARIMA (*p*,*q*,*d*) (*P*,*Q*,*D*)s is given by:

$$
\begin{aligned}
(1 - \varphi_1 B - \ldots - \varphi_p B^p)(1 - \Phi_1 B^s - \ldots - \Phi_P B^{sP})(1 - B)^d (1 - B^s)^D Y_t = \\
(1 - \theta_1 B - \ldots - \theta_q B^q)(1 - \Theta_1 B^s - \ldots - \Theta_P B^{sQ})\varepsilon_t
\end{aligned}
\tag{3}
$$

where *s* is the seasonal period, the lowercase and the capital letters represent nonseasonal and seasonal parameters, and $\varepsilon_t$ is a random variable with mean zero and the standard deviation $\sigma$.

ARIMA modeling is performed through an automated and optimized algorithm with the "auto.arima" function from the "forecast" package in R software, which is capable of traversing the space of models efficiently to arrive at the optimal model. The function detects if the training data needs a seasonal differencing, estimates unit root tests, and identifies model parameters through a step-wise approach by AICc minimization.

Artificial neural networks (ANNs) are nonparametric models that, owing to their ability to address non-linearity [58] can model complex real-world systems. An ANN is made up of nodes (i.e., its basic processing elements), links between nodes, and an activation function [59]. When past observations of a time series are introduced as inputs in an ANN structure, a feedforward neural network autoregression (NNAR) model is developed [42].

In the case of seasonal series, NNAR models take the form NNAR (*p*,*P*,*k*)*w*, where w denotes the seasonal period (here, w stands for the weekly frequency), p the nonseasonal inputs for the linear AR process, *P* stands for seasonal lags for the AR process, and *k* denotes the number of nodes in the hidden layer, such that:

$$
Y = f(H) = f(W * X + B), X = [y(t-1), y(t-2), \ldots, y(t-p)]
\tag{4}
$$

where, as in [60], *Y* is the output vector of predicted values, *f* is the activation function, and *H* = {weight matrix [(*p*\**k*)] \* input vector} + bias vector (*B*).

Feedforward autoregressive neural network models are automatically estimated by calling the "nnetar" function from R' "forecast" package. In fitting NNAR models, the nnetar function identifies parameters through 25 repetitions and AIC minimization. The number of nodes in the hidden layer is found so that *k* = (*p* + *P* + 1)/2.

More relevant details of the automatic implementation of the univariate models included in this research and the underlying theory can be retrieved from [45,55,56].

### 2.2.2. Evaluation Metrics

Several forecasting accuracy measures, both scale-dependent and scale-free (i.e., proposed by [61]) are further estimated to explore the forecasting performance of competing models over the testing window (i.e., out of sample). All can be retrieved from [45], along with more details.

Before presenting the measures, we define in Equation (5) the forecast error of a predictive model:

$$e_{T+h} = y_{T+h} - \hat{y}_{T+h|T} \tag{5}$$

where $\{y_1, \ldots, y_T\}$ represents the training set and $\{y_{T+1}, y_{T+2}, \ldots\}$ represents the test set.

$$\text{Mean error}: \; ME = mean(e_t) \tag{6}$$

$$\text{Mean absolute error}: \; MAE = mean(|e_t|) \tag{7}$$

$$\text{Root mean squared error}: \; RMSE = \sqrt{mean(e_t^2)} \tag{8}$$

$$\text{Mean percentage error}: \; MPE = mean(p_t), \text{ where } p \text{ at time } t \text{ is further given by}: \; p_t = \frac{100e_t}{y_t} \tag{9}$$

$$\text{Mean absolute scaled error}: \; MASE = mean(|q_j|),$$

$$\text{where } q_{j:} \text{ is}$$

$$q_j = \frac{e_t}{\frac{1}{N-1}\sum_{i=2}^{N}|y_t - y_{t-1}|} \text{for non-seasonal series} \tag{10}$$

$$\text{and } q_j = \frac{e_t}{\frac{1}{N-m}\sum_{i=m+1}^{N}|y_t - y_{t-m}|} \text{for seasonal time series}$$

### 2.2.3. Automatic Forecasting Technique

The robust automatic forecasting framework developed in this study encompasses the following sequential steps:

1. For each series (i.e., RSV Zoom and RSV Teams) fit all models (i.e., all specifications of ARIMA, ETS, and NNAR that are appropriate)
2. Optimize model parameters
3. Apply fitness function (i.e., AICc) to find the best model specifications in each category, for each series (i.e., best ARIMA, best ETS, best NNAR)
4. Use the best model specifications in each category (i.e., best ARIMA, best ETS, best NNAR) and the naive model to issue point forecasts over the testing period (i.e., 31-steps ahead) for each series
5. For each series, obtain accuracy measures over the testing window (out-of-sample) for predictions issued through the best model specifications in each category (ME, RMSE, MAE, MPE, MASE for best ARIMA, best ETS, best NNAR)
6. Identify best out-of-sample forecasting model for each series (i.e., Mi)
7. Re-estimate the corresponding Mi over the entire pre-pandemic window of length N = 131 for each series and optimize its parameters
8. Use optimized specifications to issue forecasts over the first pandemic months for each series (forecasting horizon spanning February 2020–August 2020 or *h* = 27 weeks)
9. Use predictions issued during step 8 as RSV baseline level for each series
10. Compute excess search interest (i.e., actual RSV level–baseline) in the aftermath of the pandemic outbreak for the two time series corresponding to the relative search interest for the two main SaaS programs.

## 3. Results and Discussion

Table 1 reports the accuracy measures for the out-of-sample forecasting ability of the four statistical and machine-learning models over the testing period. Although it emerges that there is no single model that can best predict consumer interest in videoconferencing solutions according to all accuracy measures, results nonetheless indicate that ETS and ARIMA issue superior forecasts for both Zoom and Teams RSV within the pool of competing predictive models. In particular, forecasts from ETS outperform all competing models at $h$ = 31 weeks ahead (i.e., over the pre-pandemic testing period) according to ME, MAE, and MPE for Zoom and according to all measures except for RMSE for Teams. On the contrary, the NNAR model is inferior to both ETS and ARIMA in predicting web search interest in videoconferencing apps, although it surpasses the naive forecast in some instances. However, it should be mentioned that even the over-performing model on the testing set models failed to forecast the surge in RSV for the two videoconferencing solutions during the first pandemic wave, which is first reflected by Appendix A (i.e., panel (a) for Zoom and panel (b) for Teams) and will be further revealed during the analysis of the excess RSV.

**Table 1.** Accuracy measures (out-of-sample).

| Out-of-sample forecasts for RSV Zoom | | | | |
|---|---|---|---|---|
| | ME | RMSE | MAE | MPE | MASE |
| NN | 0.51 | 0.81 | 0.71 | 9.14 | 3.92 |
| ETS | 0.48 | 0.79 | 0.70 | 8.46 | 3.88 |
| ARIMA | 0.49 | 0.79 | 0.71 | 8.50 | 3.87 |
| NAIVE | 0.59 | 0.86 | 0.74 | 10.86 | 4.07 |
| Out-of-sample forecasts for RSV Teams | | | | |
| | ME | RMSE | MAE | MPE | MASE |
| NN | 1.65 | 2.43 | 2.23 | 13.84 | 5.03 |
| ETS | −0.69 | 2.16 | 1.34 | −13.34 | 3.02 |
| ARIMA | 0.69 | 1.89 | 1.61 | 2.84 | 3.62 |
| NAIVE | 1.07 | 2.06 | 1.89 | 7.23 | 4.25 |

Ref. [62] point out that GT data might carry a sample instability issue, as the platform implements a random sampling strategy and further uses a fraction of the entire search queries to construct the RSV [43]. More recently, ref. [63] provide more information on the GT sample bias and its correction. However, ref. [64] also argue that the sampling procedure implemented by GT produces reasonably precise estimates, and consequently, a single sample suffices for data analytics purposes. An additional step has been implemented to verify the robustness of our results in this study. Consequently, estimations have been repeated on resampled RSV series. The findings were essentially unchanged, which assures the robustness of the current results.

Of note, the Diebold-Mariano test for superior forecasting accuracy that has been estimated in R software with the "dm.test" function within the "forecast" package confirms that there is a significant difference between the distribution of errors from ETS and ARIMA for forecasting Teams RSV, but cannot attest to the forecasting superiority of ETS in the case of Zoom RSV.

However, for current research purposes, we identify ETS as over-performing within the four competing models when predicting the RSV level for both Zoom and Teams over the testing period. We thus reestimate the ETS over the entire pre-pandemic dataset for each series to further issue the expected values for the respective RSV over the forecasting horizon of 27 weeks corresponding to February–August 2020, when governments started imposing pandemic-related restrictions across the globe. Next, excess web interest is quantified by extracting these baseline levels from actual RSV levels as per Equation (2).

Figure 5 offers a visual representation of the excess web-query level for Zoom and Teams over the first half of 2020.

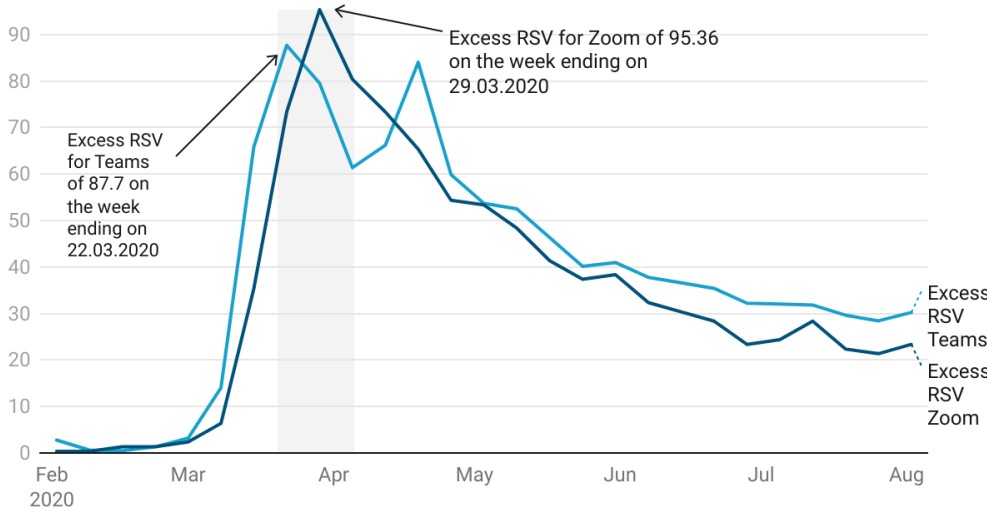

**Figure 5.** Excess search interest for the two main SaaS programs Zoom and Teams during February–August 2020. Source of data: estimation results. Chart created in Datawrapper.

## 4. Discussion

First, the current study fully backs the assertion of [29] that the use of big data such as Google Trends web queries has shifted from modeling to forecasting. Next, from a technical perspective, the forecast evaluation undertaken in this research is in line with [27] by identifying the NNAR model as the worst performer for predicting the GT RSV index. The current study also follows a similar strategy to [26] that employs Google Trends data to assess excess headache occurrences at the world level in the aftermath of the pandemic outbreak. However, different from the current results, in [26], the Holt-Winters univariate model is identified as the overperforming forecasting method for the corresponding web-search index.

Further, estimations of the excess web-search interest clearly reveal that COVID-19 has caused a surge in the web search interest for the two apps that would not have emerged in the absence of the pandemic. However, the excess customer interest registered a continuously decreasing trend after its peak by end of March 2020, suggesting on one hand an attenuation of the main impact factors that have produced the initial explosion, and on the other hand a saturation of the market for the web and videoconferencing SaaS programs. Of note, the evolution of RSV for Zoom is reflected in the decline in revenue growth from the pandemic-driven surge [65].

Another explanation could consist of the emergence of competitor applications that might have deviated the population's interest away from the two main videoconferencing solutions. However, statistics confirm that the latter argumentation does not hold, as TrustRadius reports that, as of June 2021, the market for video conferencing software is dominated by Zoom, which commands 50% of the market, followed by Microsoft Teams with 23% of market share, whereas other competitor solutions are far behind [66]. From a geographical perspective, North America is the biggest videoconferencing solutions market, due to the early adoption of emerging technologies and the existence of key market players [24]. Hence, the implementation of the breakdown of the interest by region for the video conferencing solutions in the US (reflected in Figure 6) clearly attests to the usefulness of GT data in predicting customer interest and subsequent purchases. It should be mentioned that the breakdown of the interest by region can only be accomplished at the country level, and not worldwide, which explains the choice of the biggest market for

SaaS programs to conduct this analysis. Results of this analysis thus resonate with the findings of [27,36,67], by reaffirming that web searches can be regarded as a reliant proxy for consumer shopping behavior.

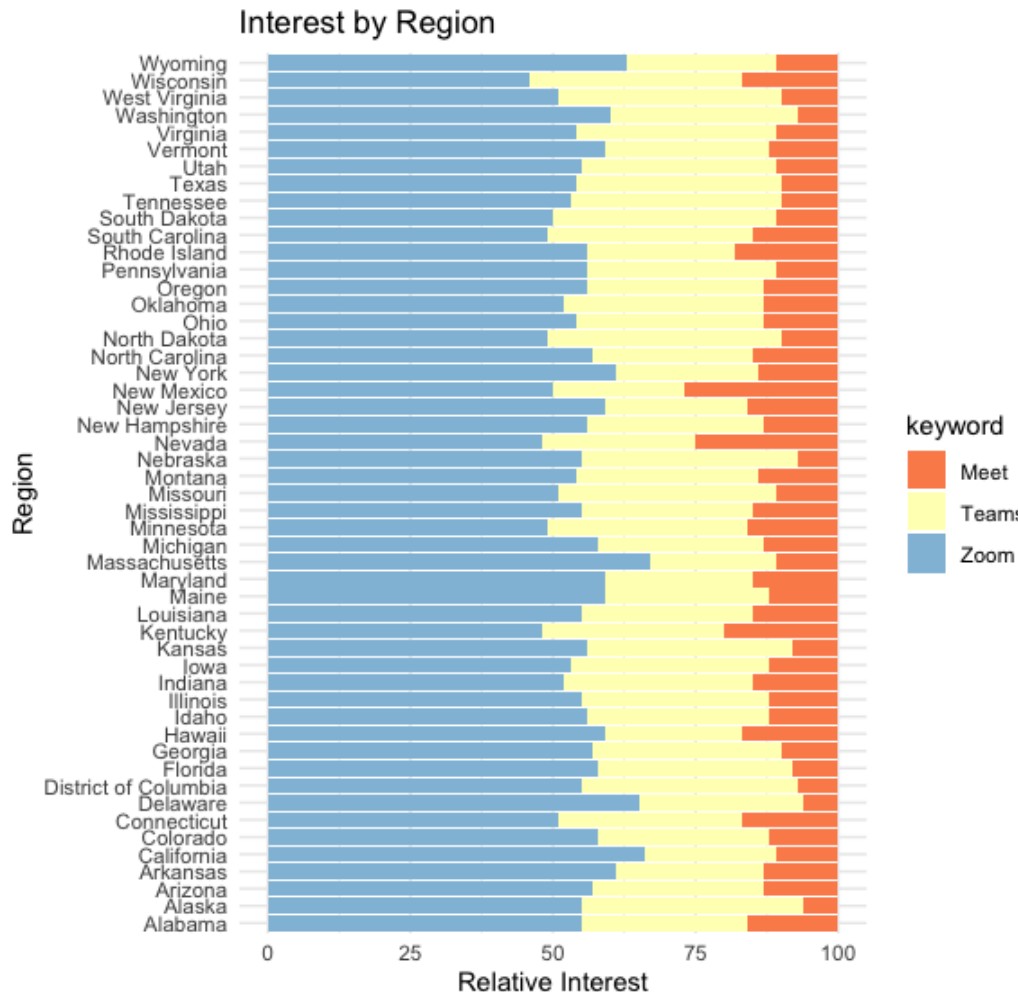

**Figure 6.** Analysis of the interest by region in the US for the top three SaaS programs (2017–2022). Created with the ggplot2 package in R environment with GT data sourced with the gtrendsR package.

Overall, research findings indicate that the global web and videoconferencing SaaS market has strongly benefited from the pandemic-driven strong demand for communication and collaboration tools. The interest for SaaS programs has stabilized at higher levels than in the pre-pandemic period, revealing a persistent impact of the COVID-19 pandemic, although much lower than during the pandemic-driven surge in interest. This resonates with [65] by indicating oversaturation of the market for SaaS programs, which further puts its main players under intense pressure to diversify their product portfolios.

## 5. Conclusions

The COVID-19 pandemic has had a multifaceted impact on societies and economies. In particular, the implementation of unprecedented government interventions aimed to reduce the spread of the pandemic has produced structural changes in industries such as IT & communications, education, and healthcare, which has, in turn, contributed to the widespread adoption of videoconferencing SaaS programs and the surge of the video conferencing market, especially its solutions segment.

Additionally, the significant progress achieved in big data analytics over the last decade has enabled the development of Google Trends (GT), a resource with huge potential for analyzing big data on worldwide online search queries [27] and forecasting consumer spending.

Consequently, GT can additionally provide relevant data to assist in the quantitative assessment of the COVID-19 impact on the video conferencing software market through the estimation of consumer excess (or abnormal) interest for two relevant SaaS programs during the early pandemic waves, which is accomplished in the current research.

This study sources reliable Google Trends web-query data by selecting the appropriate data category, compares the forecasting ability of four parametric and nonparametric forecasting models, and employs the best univariate forecasting model for "Zoom" and "Teams" Google Trends RSV to establish a baseline level for consumer interest that is finally compared to the actual web-search query level over the period February–August 2020 to produce an estimate of the excess or abnormal consumer interest in the market for videoconferencing solutions.

Results reveal that the COVID-19 pandemic has caused a surge in consumer interest for the two-videoconferencing apps that wouldn't have emerged in the absence of the pandemic. Findings also indicate that the market began losing momentum in the second half of 2020 and that excess consumer interest stabilized at a significantly lower level compared to the peak achieved at the end of March 2020, although higher than the corresponding pre-pandemic levels. Moreover, the excess consumer interest remains higher for Teams as compared to Zoom, reflecting a higher saturation of the market for Zoom.

Overall, research results coupled with the breakdown analysis by region for the US market confirm that GT RSV data can accurately reflect consumer-purchasing decisions and thus reinforce the usefulness of web search behavior for consumer spending forecasts.

Current findings have important implications for policymakers, offering relevant information to gauge new aspects of the impact and consequences of the COVID-19 pandemic. Moreover, the findings are especially informative for the main web and videoconferencing market players, revealing aggregate consumer interest and detecting main threats in the form of competition and market saturation. Future research endeavors can specifically improve forecasting based on GT data by evaluating a bigger pool of univariate and multivariate models and using data splitting rules that allow for pandemic data to be included in the training sets, thus letting the models learn from the turbulent period.

**Funding:** This research received no external funding.

**Institutional Review Board Statement:** Not applicable.

**Informed Consent Statement:** Not applicable.

**Data Availability Statement:** Data is publicly available from Google Trends.

**Conflicts of Interest:** The author declares no conflict of interest.

## Appendix A

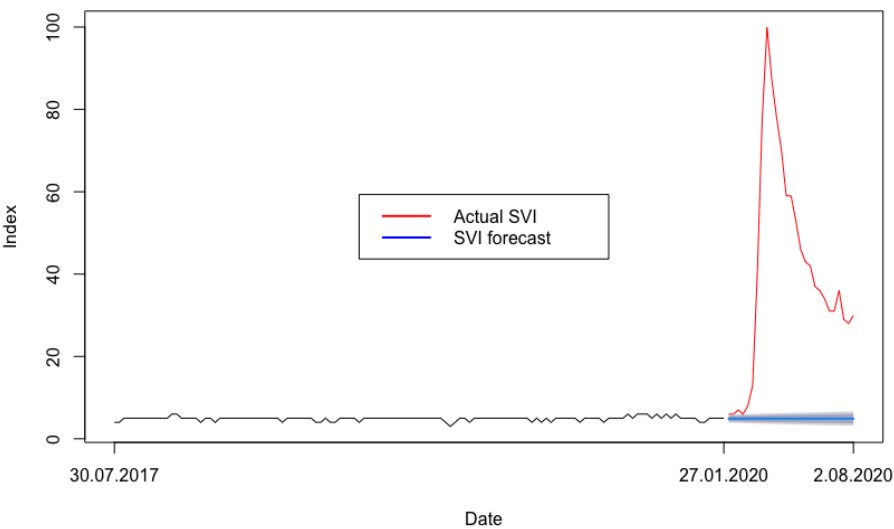

**Figure A1.** Forecasted trend for GT web queries (SVI) for the keyword "Zoom" (category "Software") at world level over 27 January 2020–2 August 2020 issued with ETS (A,N,N). Source: estimation results. Model Information: Smoothing parameters: alpha = 0.4283, Initial states: l = 4.7561, sigma: 0.444.

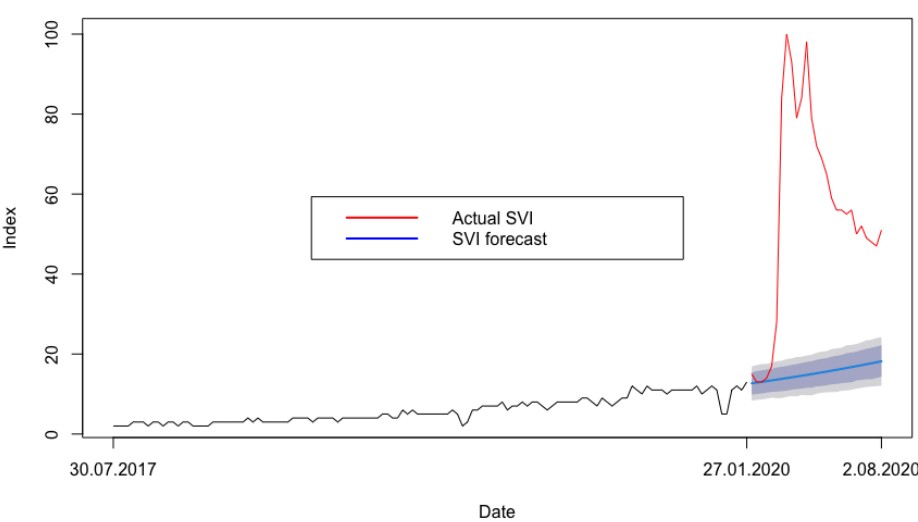

**Figure A2.** Forecasted trend for GT web queries (SVI) for the keyword "Teams" (category "Software") at world level over 27 January 2020–2 August 2020 issued with ETS (M,N,N). Source: estimation results. Model Information: Smoothing parameters: alpha = 0.0001, beta = 0.0001, Initial states: l = 2.0225, b = 1.0138, sigma: 0.1632.

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
