# Peer review of "The Impact of the COVID-19 Pandemic on the Global Web and Video Conferencing SaaS Market"

_electronics, doi:10.3390/electronics11162633_

Round 1

Reviewer 1 Report

The present article makes a study to obtain reliable Google Trends web query data by selecting the appropriate data category, comparing the predictability of four parametric and non-parametric forecasting models and employing the best univariate forecasting model for

"Zoom" and "Teams" Google Trends RSV to establish a benchmark level for consumer interest that is ultimately compared to the actual level of a web search query in the February to August 2020 period to produce an estimate of excessive interest or consumer abnormality in

the videoconferencing solutions market.

Results reveal that the COVID-19 pandemic has caused a surge in consumer interest

for the two videoconferencing applications that would not have emerged in the absence of the

pandemic. The results also indicate that the market began to lose momentum in the second mid-2020, and that excess consumer interest has stabilized at a significantly lower level compared to the peak reached at the end of March 2020, although higher than the corresponding pre-pandemic levels. In addition, excess consumer interest remains higher for Teams than Zoom, reflecting greater saturation of the Zoom market. Overall, the survey results and the detailed analysis by region for the US market confirm that GT RSV data can accurately reflect consumer purchasing decisions

and thus reinforces the usefulness of web search behavior for forecasting consumer spending. The article deals with the current topic of COVID-19, and the studies reinforce a search behavior usefulness. The article has to review its writing and see figure 2, as well as some equations that were not well defined. For example, equation (2). Equations 6-10 got a little confusing. You can use nomenclature outside of equations.

Author Response

Dear Reviewer, 
I appreciate the constructive criticism that has contributed to improving the manuscript's value. All recommendations have been implemented as follows: the indicated equations have been rewritten/redefined to improve their clarity; figure 2 has also been redone and enhanced, the text throughout the manuscript has been reviewed, and some language improvements have been made. Following other recommendations, further improvements have been performed within the introductory and results sections, and the reference list has been extended.
All modifications appear with track changes in the revised version of the manuscript. 
Sincerely,
C. Tudor

Reviewer 2 Report

1. The prediction model is too simple for deep learning because the author just used only the web search count history. I guess that if more factors like each government's decision about using video conferencing for education can be used, deep learning models like LSTM may give more correct results.

2. In Section 3, the author gave the average results. However, she needs to give an example of predictions of the three models to help the reader's understanding.

3. The author should give a related work section.

Round 2

Reviewer 2 Report

The author gave proper responses.